# Development of the Peritoneal Metastasis: A Review of Back-Grounds, Mechanisms, Treatments and Prospects

**DOI:** 10.3390/jcm12010103

**Published:** 2022-12-23

**Authors:** Kaijie Ren, Xin Xie, Tianhao Min, Tuanhe Sun, Haonan Wang, Yong Zhang, Chengxue Dang, Hao Zhang

**Affiliations:** 1Department of Surgical Oncology, The First Affiliated Hospital of Xi’an Jiaotong University, Xi’an 710061, China; 2Department of Nuclear Medicine, The First Affiliated Hospital of Xi’an Jiaotong University, Xi’an 710061, China

**Keywords:** peritoneal carcinomatosis, molecular mechanisms, treatment

## Abstract

Peritoneal metastasis is a malignant disease which originated from several gastrointestinal and gynecological carcinomas and has been leading to a suffering condition in patients for decades. Currently, as people have gradually become more aware of the severity of peritoneal carcinomatosis, new molecular mechanisms for targeting and new treatments have been proposed. However, due to the uncertainty of influencing factors involved and a lack of a standardized procedure for this treatment, as well as a need for more clinical data for specific evaluation, more research is needed, both for preventing and treating. We aim to summarize backgrounds, mechanisms and treatments in this area and conclude limitations or new aspects for treatments.

## 1. Introduction

Peritoneal metastasis refers to the development and spread of several gastrointestinal and gynecological carcinomas in the peritoneal cavity; related carcinomas include colorectal carcinoma, gastric carcinoma, ovarian carcinoma, etc. [1]. There are other primary peritoneal tumors, which mostly originated from mesothelium, such as serous carcinoma, malignant mesothelioma and pseudomyxoma peritonei. Nearly all types are similar in presentation, diagnosis, evaluation and treatment [2,3].

The occurrence of peritoneal metastasis has been indicated to significantly decrease overall survival in patients with gastrointestinal cancer. Due to the lack of effective systemic chemotherapy, peritoneal metastasis is mostly considered a terminal condition. A study from France collected data from 1976 to 2012 for 9148 patients with colorectal adenocarcinoma and indicated that the proportions of patients who underwent curative resection for synchronous and metachronous peritoneal carcinomatosis were 11% and 9%, respectively, and these patients had 3-year overall survival rates of 8% and 5% [4]. Additionally, patients with peritoneal metastasis are more likely to relapse than those with single tumors, regardless of what methods were used during treatment [5]; however, active treatment still prolongs overall survival [6].

In the last century, peritoneal metastasis was treated as a terminal and uncurable condition, but treatment advances, especially cytoreductive surgery (CRS) and hyperthermic intraperitoneal chemotherapy (HIPEC), have improved the life quality of patients; thus, the outcomes of peritoneal carcinomatosis have been changed [7]. Currently, several new methods, such as pressurized intraperitoneal aerosol chemotherapy, new delivery systems targeting local regions and molecular targeted drugs, have been proposed. A few phase II and phase III studies have investigated the effects of multimodal approaches and are still ongoing. The aim of this review was to acknowledge widely accepted concepts of peritoneal carcinomatosis, summarize the physiology and pathophysiology of this disease and assess new treatments for peritoneal carcinomatosis to have a better understanding of this condition.

## 2. Physiology and Function of the Peritoneum

The peritoneum, as a serous membrane, consists of a monolayer of mesothelial cells connected to a basement membrane on a layer of connective tissue. The submesothelial layer consists of the extracellular matrix, which is composed of multiple types of collagens, glycoproteins, glycosaminoglycans and proteoglycans. Mesothelial cells are present in a single layer and generally show a flattened, stretched, squamous or cuboidal appearance. Different types of mesothelial cells are present at different positions and show various functions. Mesothelial cells with cuboidal shapes are found mostly at the visceral peritoneum; compared to cells with other shapes, these cells contain rougher endoplasmic reticulum and Golgi apparatus organelles, microtubules, intracellular vesicles and microfilaments, indicating their good biosynthetic capacity and active transmembrane transport. Vascular structures and lymphatic systems are present in the subserous space [8].

The peritoneum allows the exchange of molecules and the production and transportation of peritoneal fluid, which provides a suitable environment for intra-abdominal organs. Recently, other functions of the peritoneum have been found, including glycosaminoglycan and surfactant secretion, inflammation inhibition, leucocyte migration, antigen presentation and tissue repair. Functional peritoneal cells are vital in maintaining peritoneum homeostasis. Its role in tumor dissemination has also been studied due to the secretion of growth factors and the presence of specific structures, such as milky spots [9].

Milky spots, which are submesothelial lymphoid structures, are found in the peritoneum, especially in the omentum. Milky spots consist of aggregates of mesenchymal cells surrounding capillary convolutions, mostly macrophages, lymphocytes, type 2 innate lymphoid cells (ILC2s) and CXCL13^+^/FDCM1^+^ stromal cells [10]. In recent studies, milky spots were found to be linked to tumor metastasis, indicating that more attention should be given to peritoneal carcinomatosis.

## 3. Peritoneal Metastasis

Peritoneal metastasis occurs via multiple ways. Usually, gastrointestinal (bowel wall penetration in cases of colorectal cancer) and gynecological carcinomas could directly implant cancer cells into the peritoneum, or via lymphatic ways, either due to a full-thickness invasion of the bowel wall of a primary tumor or blood and lymphatic vessels damage during a procedure. Other types of cancer, such as lung cancer, could occur through blood flow [1,11,12].

There are widely accepted steps in peritoneal metastasis. First, single cancer cells or clumps detach from the primary tumor and spread into the peritoneal cavity. Second, these cancer cells become susceptible to regular peritoneal transport along predictable routes. Third, these cells attach to and invade the distant peritoneum. Fourth, the cells invade the subperitoneal space. The underlying connective tissue provides the necessary scaffold for tumor proliferation. The final step involves angiogenesis, which sustains tumor proliferation and enables further growth [11] (Figure 1).

Single cancer cells or clumps detach from the primary tumor, spread into the peritoneal cavity and attach to the distant peritoneum; then, cancer cells invade the underlying connective tissue in different ways.

The detachment of tumor cells from the primary tumor could be spontaneous or secondary due to internal and external factors (mostly due to improper iatrogenic operation) [13]. Usually, high interstitial fluid pressure leads tumor cells to detach spontaneously due to rapid cell proliferation, high tumor blood vascular permeability, lack of functional lymphatic circulation, interstitial fibrosis and contraction of the interstitial matrix mediated by stromal fibroblasts [14]. However, it is interesting that performing laparoscopy did not seem to increase the risks occurring with peritoneal metastasis, leaving choices which are better for the outcomes of patients [15].

Once tumor cells detach from the primary tumor, they are mostly transplanted into four regions: the Pouch of Douglas at the rectosigmoid level, the right lower quadrant at the lower end of the small bowel mesentery, the left lower quadrant along the superior border of the sigmoid mesocolon and colon and the right paracolic gutter lateral to the cecum and ascending colon. The interaction of gravity, diaphragmatic excursion, mesenteric reflections and peritoneal recesses leads to flowing toward the pelvis and from the pelvis, along the right paracolic gutter, toward the subdiaphragmatic space [11].

It is worth noting that once tumor cells detach from the primary location, they must survive before they successfully metastases. Once cancer cells are in the peritoneal cavity, a change of the environment stimulated tumor cells, which favored their survival. Anoikis resistance is the most related phenotype that cancer cells develop which grants their ability to survive in the peritoneal cavity. There are ways tumor cells escape death and develop anoikis resistance, including modifying surface molecules, regulation of activating pathways and other mechanisms. For example, several transcription factors and genes are activated in gastric cancer cells; nuclear MTH9-VTNNB1, TCF7L2-PLAUR and NOX4-EGFR/ROS could promote anoikis resistance [16,17,18]. Another mechanism is that the activation of Integrins have been found to be participating in metastasis in ovarian cancer; integrin α5β3 sustained cell survival and resisted anoikis by activating transcription of Bcl-2 [19]. The integrins also protected gastric cancer cells from anoikis as well [20]. Furthermore, upregulating KRAS and MEK-ERK in ovarian cancer cells could stabilize spheroid formation; in this form, it has advantages in promoting survival and metastasis than single cells, and tumor-associated macrophages also involved in spheroid formation, by promoting binding of cancer cells and activating EGFR signaling pathways [21,22]. In all, cancer cells must achieve certain molecular changes so that they can survive once they detach from the primary tumor, which explains why a high expression of some molecules, for example, CXCR4 in gastric cancer, have a higher rate of peritoneal metastasis occurring [23].

Peritoneal cells are essential in occurring metastasis by interacting with cancer cells in multiple ways: direct physical contact, soluble cytokines in microenvironment regulation by paracrine activities or reaction with a matrix, such as ECM components. There are mainly several mechanisms in peritoneal metastasis. Several types of cancer cells could induce apoptosis of peritoneal mesothelial cells, causing a breach onto the peritoneum, which will be mentioned below. Usually, an inflammatory environment would cause peritoneal cells to produce cytokines and adhesion molecules, resulting in peritoneal metastasis. TGF-beta secreted by gastric cancer cells was studied to induce changes of mesothelial cell morphology, leading to peritoneal fibrosis, or changing into fibroblast phenotype, ending in peritoneal metastasis [24,25,26].

Senescent peritoneal mesothelial cells are found to promote peritoneal metastasis. SASP, senescence-associated secretory phenotype, refers to several highly secreted cytokines (e.g., IL-1, IL-6, IL-13, IL-15), chemokines (e.g., CXCL1, CXCL8, CXCL12, ENA-78), growth factors (e.g., EGF, FGF, HGF, TGF-β, VEGF, angiogenin, epiregulin), extracellular matrix (ECM) proteins and remodeling enzymes (e.g., MMP-1, -3, -10), soluble receptors and ligands (e.g., ICAM-1, EGF-R, Fas) and other elements that promote cancer proliferation and metastases [27].

Malignant ascites is a lipid-rich microenvironment, mostly from differentiated preadipocytes stimulated by cancer cells. These cells mature and release free fatty acids, therefore enhancing cancer cell proliferation and EMT. Cancer cells in ascites also tend to produce more enzymes from fatty acid metabolism, with enhanced fatty acid oxidation, promoting peritoneal metastasis [28,29].

Adhesion and invasion into the peritoneum involve multiple molecular mechanisms facilitating cancer cell metastasis (Figure 2). These cells could directly adhere to the peritoneum and invade the submesothelial tissue, or cells could travel through milky spots. Although the immune function of milky spots is important for the defense against microorganisms, they are involved in a pathway for peritoneal carcinomatosis [30]. Milky spots usually provide a microenvironment suitable for cancer cells to implant and grow. Several studies have investigated how gastric cancer cells are implanted and transferred through milky spots: they provide a hypoxic microenvironment for gastric cancer cells to implant and grow, and HIF-1α in the microenvironment plays a significant role during progression. Hypoxia could also induce the EMT of gastric cancer cells [31]. Macrophages in milky spots produce MDC/CCL22 and its receptor CCR4, which are highly expressed in the omentum and the diaphragm underlining, contributing to gastric cancer cell survival and growth [32].

### 3.1. Adhesion to the Peritoneum

Once tumor cells detach from the primary tumor, they float in the peritoneal fluid and travel until they contact the peritoneum. Adhesion of free cancer cells to the peritoneal surface relies on several adhesion molecular mechanisms, such as several integrins, proteoglycans and the immunoglobulin superfamily.

#### 3.1.1. Immunoglobulin Superfamily

The immunoglobulin superfamily is a group of cell adhesion molecules, including ICAM-1, VCAM-1 and L1CAM. Intercellular adhesion molecule-1 (ICAM-1) is a surface molecule expressed by mesothelial cells, cancer cells and endothelial cells [33]. It was found that it could enhance tumor cell adhesion mediated by IL-6 or TNF-α [33]. However, Hiroaki et al. indicated that ICAM-1 could reduce lymph node metastasis, which left the specific effect of ICAM-1 in peritoneal metastasis unknown [34].

Vascular cell adhesion molecule-1 (VCAM-1) is a known, highly expressed membrane protein on mesothelial cells and has been found to play a role in adhesion to the peritoneum in ovarian cancer [35]. A current study found that VCAM-1 was related to clinicopathological factors in colorectal cancer, such as lymph node metastasis and clinical stage [36], suggesting that it might play a role in multiple types of cancer.

L1 cell adhesion molecule (L1CAM), an important molecule and marker found in ovarian cancer for poor prognosis, has been found to be related to adhesion and invasion, and an antibody against L1CAM could significantly suppress this progression [37]. It is also involved in the metastatic process in gastric cancer, predicting metastasis-related clinicopathological features and unfavorable outcomes, and could be a feasible predictor of oncological outcome [38].

Studies have found that nectin-2, an adhesion molecule participating in cell proliferation, differentiation and migration of epithelial, endothelial, immune and nervous cells, is associated with tumor growth, adhesion and angiogenesis in ovarian cancer [39]. It was found to be significantly upregulated in patients who had lymph node metastasis or residual tumors >1 cm after surgery, as well as in samples of tumor tissues and lesions on the peritoneum, which suggest its role in metastasis of ovarian cancer [40].

#### 3.1.2. Proteoglycans

CD44, a cell-surface proteoglycan, participates in behaviors such as cell interaction, adhesion and migration. It is overexpressed in gastrointestinal and gynecological cancers [41]. Specifically, several studies found that CD44 partly mediates adhesion, such as that exhibited by cancer cells attaching to peritoneal mesothelial cells [42]. CD44-mediated adhesion could also partly explain metastasis in the inflammatory microenvironment after surgery, in which several cytokines, such as TGF-β1, IL-1b and TNF-α, are generated, resulting in upregulated CD44 expression [43].

#### 3.1.3. Integrins

Integrins are a superfamily of cell adhesion receptors consisting of 24 members, each of which is composed of α and β subunits, and recently, integrins participating in cancer metastasis have been investigated [44]. Studies have found that integrin α2β1 participates in the peritoneal metastasis. Furthermore, it is a potential target for the treatment of peritoneal metastasis [45]. Integrin α3β1 was also found to be involved in the adherence of gastric cancer cells to the peritoneum [46]. Integrin α4β1 partly mediated peritoneal metastasis of ovarian cancer; furthermore, antibodies against it could increase ovarian cancer response to carboplatin, while treating with antibodies alone showed no response [47]. This phenomenon shows potential in clinical use.

Due to the hypoxic microenvironment, SIRT1 is degraded via the autophagic lysosomal pathway, causing increased acetylation of HIF-1α and secretion of VEGFA. Under these conditions, VEGFA derived from peritoneal mesothelial cells acts on VEGFR1 in gastric cancer cells, increasing integrin α5 and fibronectin expression, causing further adhesion to the peritoneum [48].

#### 3.1.4. CXC Subfamily

SDF-1α is a chemokine of the CXC subfamily on mesothelial cells. Its upregulation was indicated to possibly be due to bioactive cytokines secreted from tumor cells and was found to be associated with enhanced intraperitoneal dissemination of epithelial ovarian carcinoma cells. Another possible mechanism is that CXCR2 secreted by CT-26 colon cancer cells could induce cell proliferation and migration by combining with CXCL2 on ECM components, blocking this process inhibited cell proliferation and migration [49].

#### 3.1.5. Other Molecules

Wnt5a is a noncanonical Wnt ligand that is highly expressed in ascites in female patients with ovarian cancer and promotes ovarian tumor cell adhesion, migration and invasion. The downstream effector is the Src family kinase Fgr, which is a potential target for the treatment of ovarian cancer [50]. Other molecules are being investigated for possible treatment.

Currently, there are various molecules in research connecting adhesion to peritoneum, and several of them showed the possibilities of predicting outcomes or providing treatment opinions. Whether those mechanisms could be used in vivo, and their effect, is still in need of investigation.

### 3.2. Invasion into the Peritoneum

After adhesion, tumor cells need to invade the submesothelial tissue to achieve colonization; this process could be adhering to the ECM through the gap between mesothelial cells or directly induce mesothelial cell apoptosis. Carbon dioxide pneumoperitoneum temporarily enlarges intercellular clefts and exposes the ECM, allowing tumor cells to access the ECM more easily by using RGD peptides or pseudo-RGD peptides [51]. Tumor cells could also directly influence the function of mesothelial cells. Heath et al. found that SW480 colorectal cancer cells could induce FAS-dependent apoptosis of cultured human mesothelial cells and that tumor–mesothelial adhesion was essential for inducing apoptosis. This study suggested that this phenomenon plays a role in peritoneal carcinomatosis [52].

Several studies have shown that matrix metalloproteinases (MMPs) contribute to the invasion of the submesothelial tissue by causing degradation of the ECM and contraction of mesothelial cells. MMPs are a family of zinc-dependent endopeptidases that are involved in the degradation of various proteins in the extracellular matrix (ECM). Their functions in cancer invasion and metastasis have been found gradually. MMP-7 is likely to be associated with adhesion, as the downregulation of MMP-7 could suppress invasion without influencing proliferation; it also takes part in serosal involvement, lymph node metastasis, poor differentiation of cancer and peritoneal dissemination, indicating its role in peritoneal adhesion [53]. Another matrix metalloproteinase is MMP-9. In vitro studies have shown that peritoneal mesothelial cells can also secrete MMP-9 under TNF-α stimulation in gastric cancer cells, which enhances cancer cell invasion [54]. MMP2/9 were found strongly upregulated in colon tumor tissues, and inhibition of them could reduce colonization [55].

## 4. Diagnosis and Evaluation of Peritoneal Metastasis

The diagnosis of peritoneal carcinomatosis relies on imaging, biopsy and laparoscopy. CT and PET/CT are the most used methods for the detection of peritoneal metastasis; an enhanced CT scan could provide valuable information for metastasis detection [56], especially for pseudomyxoma peritonei [57]. PET/CT with radioisotopes is more sensitive for diagnosis than CT, according to Koh et al. [58]; however, other research found CT to be more sensitive than PET/CT for diagnosis [59]. PET is traditionally more sensitive to tumors with hypermetabolic uptake, but not minor nodules [58,60]. Diffusion-weighted (DW) MRI was another method and seemed to be the same as CT in sensitivity or PET/CT in diagnosis [61,62]. In addition, PET/CT appeared favorable in sensitivity as well, but showed weak ability in excluding diagnosis of peritoneal metastasis [63,64]. Imaging methods can assist in assessing peritoneal metastasis, thus evaluating if cytoreduction is possible. Compared with imaging, the most precise method of diagnosis is peritoneal visualization and biopsy, for example, exploratory laparotomy, but this approach is invasive [65]. It is worth noting that, recently, a fluorescent probe called gGlu-HMRG had been used to detect tiny tumors on the peritoneal wall, showing potential in both diagnosis and assistance for fluorescence-guided surgery for peritoneal carcinomatosis [66]. Currently, despite the limitations of CT, it is a still powerful and cost-effective tool for general metastasis detection, making it the first choice for detection and diagnosis; PET/CT and MRI could be used in an alternative way and in specific situations.

A way of evaluating peritoneal metastasis is using the peritoneal cancer index (PCI). PCI includes the surgical peritoneal cancer index (sPCI) and pathologic peritoneal index (pPCI), the former of which requires evaluating peritoneal metastasis during surgery. Surgeons record the number and size of lesions in each of the 13 peritoneal regions and add them to obtain the sPCI. The pPCI is scored through the pathologic evaluation of specimens. sPCI and pPCI do not always seem consistent, for the former is mostly subjective, though sPCI could provide valuable information for the evaluation of patients [67]. Though pPCI is more objective, specimens would shrink during the process, causing misjudgment of tumor size. Furthermore, there were no standard procedures for the evaluation of specimens from CRS, thus, further research needs to be conducted [68]. CT-PCI used a CT scan for the evaluation of the disease burden and prognosis, helping for these aspects, regardless of its accuracy [60,69].

It is also interesting to evaluate whether there were differences between primary tumors and metastasis tumors in molecular and gene expression to further understand the mechanisms of metastasis and to provide targeted treatment. Several studies found high consistency in colorectal cancer in both dMMR, MSI status and biomarkers [70,71]; however, there were other studies that found different expression in colorectal cancer between primary tumor and metastases, a significant enrichment for CMS4 in peritoneal metastasis, providing a possible treatment combined with CRS-HIPEC to reduce metastasis tumors [72]. Furthermore, different expressions or mutations were detected in gastric cancer based on a multi-omic profiling, suggesting a molecular-targeting therapy separate from therapy on primary tumors.

The differences of biomarkers between colorectal cancer and its metastases have been compared [73,74], including KRAS/BRAF mutation and MSI status, indicating the importance of testing mutations in peritoneal metastasis and treating methods. However, there is still in lack of research and data in the area, which suggests a further evaluation of personalized treatment.

## 5. Treatment to Peritoneal Metastasis

Peritoneal metastasis was traditionally considered a terminal condition and thus lacked effective treatment. However, several methods and ideas have been proposed and used in the clinic with exciting progress (Figure 3).

In surgery for primary tumors, traditional ways to reduce the incidence of peritoneal metastasis are to follow a no-touch isolation technique (NTIT)—complete removal of adjacent invaded structures and surgical margins deep in the healthy tissue—and other standard surgery procedures, which reduce the feasibility of surgery-induced local metastasis and blood metastasis [75]. However, recent clinical trials questioned the superiority of the NTIT [76], indicating that further treatments are needed.

Intraperitoneal chemotherapy has been widely applied, compared to systemic chemotherapy for peritoneal carcinomatosis, because not all reagents of systemically applied chemotherapy could be fully delivered to the peritoneum, possibly due to the peritoneum-plasma barrier. This barrier leads to peritoneal clearance being much slower than systemic clearance; thus, a high intraperitoneal chemotherapy dose would result in moderate systemic drug exposure [77].

HIPEC, known as hyperthermic intraperitoneal chemotherapy, uses specific chemical reagents and a high temperature to kill tumor cells. It has mainly been evaluated in peritoneal carcinomatosis in colorectal, mucinous appendicular adenocarcinoma and ovarian cancer [78,79,80,81,82,83]. The main advantage of HIPEC is the maintenance of a high regional reagent concentration, and blood drainage of the peritoneal surface occurs via the portal vein to the liver. Increasing the concentration in the liver would suppress liver metastasis as well. Another advantage is that 41–43 °C hyperthermia could directly kill tumor cells by inhibiting RNA synthesis and mitotic arrest and increasing the number of lysosomes and the activity of lysosomal enzymes. Heat also increases the cytotoxicity of certain chemotherapeutic drugs and enhances tissue penetration [84]. HIPEC is usually administered in the operating room immediately after CRS, due to limited drug penetration in tumor tissues, and is mainly used to kill microscopic residual disease after CRS.

However, due to the direct administration of drugs into the peritoneum, choices of these drugs must meet certain criteria. Typically, these reagents should be effective against malignant cells and low local toxicity after administration. Additionally, these reagents should be cycle-nonspecific and induce heat-synergized effects [84]. Specific reagent effects, toxicity to malignant cells and penetration into tumor tissues during systemic chemotherapy should be considered to determine which method is more effective. Reagents that require transformation into an active form in the liver should be excluded. In addition, the most important feature is that reagents should be slowly absorbed from the peritoneal cavity and rapidly cleared via hepatic and/or renal mechanisms so that a high concentration of drug can be maintained with low systemic toxicity [85].

Several studies have compared factors influencing the outcomes of CRS-HIPEC (Table 1). Though most studies used different strategies, the results suggested that a patient’s survival was prolonged after a complete procedure. Factors involved in HIPEC include choices of drugs, applied dose, duration, carrier solution, perfusate volume, perfusate concentration and use of an open vs. closed technique [85]. Interestingly, repeated CRS-HIPECs seemed to be beneficial for patients occurring metastasis limited to peritoneum, suggesting that it might be suitable for specific patients to prolong survival [86]. However, HIPEC had risks of causing changes to genetic patterns between tumors and normal tissues and an upregulation of heat shock-related genes, to be specific, which would be an adverse effect, and an idea of combining other treatments [87].

Overall, most studies recommended that CRS-HIPEC could improve outcomes, with the restriction that low disease extent (limited peritoneal metastasis) and complete CRS indicated better outcomes, indicating its limitation in clinical use [88]. However, there is still a lack of researchers comparing different strategies and different factors of HIPEC, leading to uncertainty for specific treatment efficiency. Therefore, further research and new methods should be proposed and the advantages of different strategies for a limited number of procedures for different stages of peritoneal carcinomatosis, due to disadvantages, complications and limitations for CRS-HIPEC, should be compared, and personalized treatments should be provided in the future.

A new method called pressurized intraperitoneal aerosol chemotherapy (PIPAC) has been studied and brought to clinical use. Compared with systemic chemotherapy and chemical solutions administered to the peritoneal cavity, PIPAC could optimize the uniformity of chemical concentrations in the peritoneal cavity, enhance drug penetration by increasing intraperitoneal hydrostatic pressure against interstitial fluid pressure, limit blood outflow and adjust the environment of the peritoneal cavity, such as temperature and pH, to achieve better tissue targeting [89]. A cohort study investigated PIPAC combined with systemic chemotherapy to treat diffuse malignant peritoneal mesothelioma (DMPM). In this study, 26 patients with unresectable disease were treated with PIPAC, and 20 of them had not previously received CRS. An improvement of symptoms was reported for 32% of the patients, and control of ascites was reported in 46%. Fourteen of fifteen patients were treated with CRS plus HIPEC and achieved complete resection. The median overall survival period was 12 months, and the median progression-free survival (PFS) was significantly better among the patients who underwent resection than among those who did not (33.5 vs. 7.4 months, *p* < 0.001). This study demonstrated that for patients with unresectable DMPM, PIPAC could be used as neoadjuvant chemotherapy and increase the possibility for further CRS [90]. Alyami et al. found that unresectable peritoneal metastasis treated with repeated PIPAC could allow secondary treatment: CRS- HIPEC [91]. There are multiple studies which have evaluated the safety and feasibility of PIPAC combined with chemical drugs, but oncological outcomes required more evaluation [92,93]. Compared with CRS and HIPEC, PIPAC is more suitable for peritoneal metastases of various origins that cannot be treated by resection. After repeating PIPAC, some cases of unresectable disease could be treated via secondary resection.

There are new methods, such as electrostatic precipitation pressurized intraperitoneal aerosol chemotherapy (ePIPAC), which use electrostatic precipitation of aerosols to achieve stronger penetration and more even distribution [94]. Studies have tested safety and tolerance in treated patients, but efficiency was debatable [95,96]. Another new method, hyperthermic pressurized intraperitoneal aerosol chemotherapy (hPIPAC), which involves the application of cisplatin at temperatures of 38.8–40.2 °C [97], has been proposed and tested recently. Both require further experiments to evaluate feasibility and long-term therapeutic effect.

High-intensity ultrasound (HIUS) has been studied to treat several solid tumors, and the purpose of HIUS was to further enhance tissue penetration, which has been reported [98]. The damage HIUS could cause has also been assessed, and it could yield measurable microscopic changes on the peritoneal surface with minimal damage [99]; however, as a new theory, specifics regarding its usage, safety and combination with other methods, such as CRS plus HIPEC, PIPAC or new biocompatible materials, should be further assessed.

Neoadjuvant intraperitoneal and systemic chemotherapy (NIPS) is a new method aiming to increase possibilities to access CRS, especially for those whose tumor features are not suitable for CRS. A meta-analysis was performed on 8 retrospective studies, including 373 patients with peritoneal metastasis from gastric cancer, 109 of whom continued NIPS treatment because of macroscopic peritoneal metastasis and 265 of whom received surgery for no macroscopic peritoneal metastasis. NIPS combined with surgery significantly improved survival compared to those without surgery, and NIPS could increase the possibility of achieving R0 resection [100]. Other studies supported the idea that NIPS could be used for advanced gastric cancer with peritoneal metastasis [101]. Due to a lack of further clinical data, more clinical trials and research should be conducted to confirm and evaluate this hypothesis.

Drugs targeting adhesion molecules and immunotherapies have shown potential in preventing peritoneal metastasis. Zang et al. found that LPPR4 (which plays a role in promoting peritoneal metastasis of gastric cancer through Sp1/integrin α/FAK signaling) could be a new therapeutic target [102]. The binding of CXCL12 to CXCR4 and CXCR7 on tumor cells leads to antiapoptotic signaling through Bcl-2 and Survivin upregulation; it also promotes EMT through the Rho-ROCK pathway and leads to alterations in cell adhesion molecules. AMD3100 (Plerixafor or Mozobil) is a small molecule CXCR4 antagonist used in clinical trials for gastrointestinal tumors [103] and shows potential prospect. Methods activating immune effects of anti–tumors were investigated due to the high expression of PD-L1 during the process of peritoneal metastasis [104]. CMP-001, a virus-like particle composed of the Qβ bacteriophage capsid protein, encapsulating a CpG-A oligodeoxynucleotide, could activate lasmacytoid dendritic cells and interferon alpha release [105], which might contribute to an anti-tumor response by the development of T-cells [106]. Similarly, oncolytic virotherapy was also classified as another type of immunotherapy; this therapy used viruses as oncolytic vector platforms for the delivery of different treatment agents, such as therapeutic genes, prodrug convertases, toxins, sodium iodide symporter for radiotherapy and immunomodulators [107,108]. JX-594 (pexastimogene devacirepvec, Pexa-vec) is an oncolytic vaccinia virus armed with GM-CSF; a murine version of it shows potential as an anti-tumor by activating dendritic cells and CD8 T cells to enhance their infiltration into peritoneal tumor nodules. Furthermore, it could combine with immune checkpoint inhibitors to induce enhanced immunity to kill metastases [109].

Localized chemotherapy could decrease the toxicity of chemical drugs systemically and maintain a higher concentration in specific areas. In addition to HIPEC and PIPAC, new delivery systems are being studied. Biocompatible carrier systems, such as hydrogels, cells and peptides, have been used for localized drug delivery for the treatment of peritoneal metastasis.

Hydrogels are 3D networks of crosslinked hydrophilic polymer chains that can be formed by different materials and show various abilities. Hydrogels designed for different situations could be sensitive to pH, temperature and physical stimuli, such as light or UV, which means they could protect contents from extreme environments and deliver them in certain areas [110]. Several delivery systems based on hydrogels have shown feasibility in treating peritoneal metastasis (Table 2) [111,112,113,114].

Another delivery system utilizes cells as carriers. Ling et al. used engineered doxorubicin-loaded M1 macrophages (M1-Dox), which overexpress CCR2 and CCR4, to target cancer cells; M1-Dox transferred drug cargoes into tumor cells via a tunneling nanotube pathway. The results showed that delivering drugs (Dox) from cell to cell was more efficient than lysosomal delivery in terms of effective concentration and drug loading. Furthermore, these cells were not only effective in treating primary tumors, but also had a great advantage in treating metastasis [115]. Functional amyloids produced in bacteria as nanoscale inclusion bodies are a new pathway for treatment. Céspedes et al. used Pseudomonas exotoxin (PE24)-formed bacterial inclusion bodies functionalized with CXCR4 and found that colorectal cancer mouse models treated with these proteins showed significant arrest of tumor growth without toxicity [116]. Albumin, with multiple cellular receptor and ligand binding sites, which are able to bind and transport numerous endogenous and exogenous compounds, could also act as a carrier for chemotherapy drugs targeting peritoneal metastasis, providing a more biocompatible approach for drug delivery [117].

## 6. Conclusions

Overall, peritoneal metastasis is usually considered a terminal condition in patients and could be derived from several gastrointestinal and gynecological carcinomas. Although an increasing number of molecules have been found to be involved in peritoneal metastasis, the mechanisms of peritoneal metastasis are still complicated, and effective ways to treat them synchronously are lacking. Thus, the specific mechanisms of early tumor cell transfer at the gene and molecular levels should be studied. Recent research on the treatment of peritoneal metastasis has mainly focused on CRS, HIPEC, PIPAC and surgery combined with chemotherapy to local regions. Further studies are needed regarding new methods for enhancing tumor penetration, increasing local drug concentrations, decreasing toxicity and regarding better solutions for patients with advanced peritoneal metastases.

CT is currently the first choice for diagnosis and, combined with MRI or PET/CT, could be more accurate. New materials and methods, such as fluorescence probes, should be proposed for the detection of early minor metastasis so that timely treatment could be taken to prevent further progress.

The tumor microenvironment and interaction between tumor cells and other cells of the peritoneum could be potential targets for treatment. Traditional treatment strategies using chemical drugs need to be improved and new methods need to be created or combined with traditional methods. Furthermore, personalized treatment and health care for patients should also be considered.

## Figures and Tables

**Figure 1 jcm-12-00103-f001:**
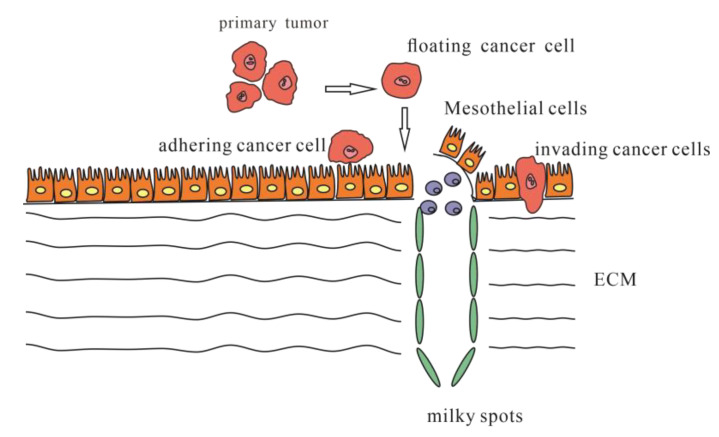
Steps of occurring peritoneal metastasis.

**Figure 2 jcm-12-00103-f002:**
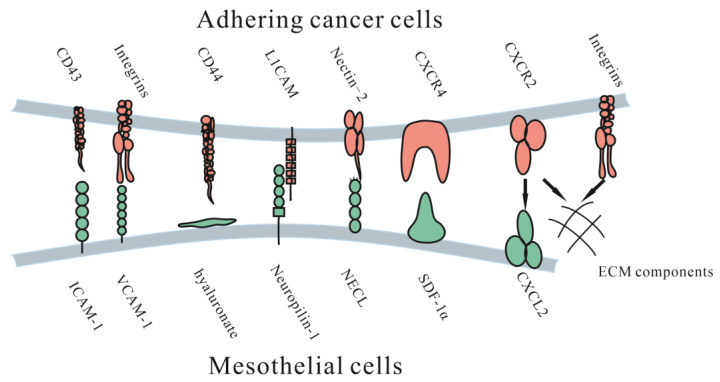
Molecular mechanisms of adhesive interactions mediating peritoneal carcinomatosis. ICAM-1 = intercellular adhesion molecule-1. VCAM-1= vascular cell adhesion molecule-1. L1CAM = L1 cell adhesion molecule. NECL = Nectin-like (NECL) family. CD44 = hyalonurate receptor. SDF-1α = stromal cell-derived factor 1α. CXCR4 = CXC receptor 4. CXCR2 = CXC receptor 2. CXCL12 = CXC ligand 12.

**Figure 3 jcm-12-00103-f003:**
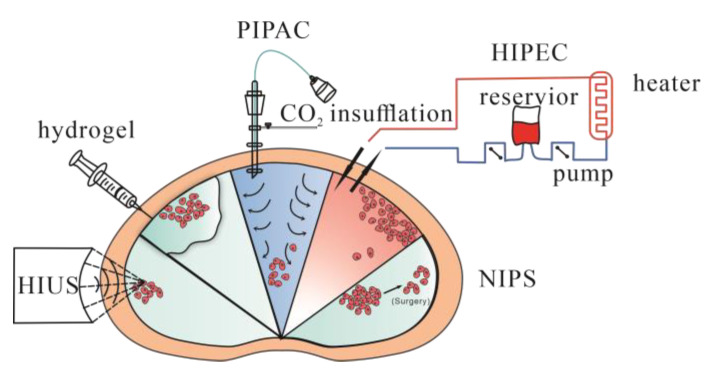
Treatments being applied to peritoneal metastasis. HIPEC = hyperthermic intraperitoneal chemotherapy. PIPAC = pressurized intraperitoneal aerosol chemotherapy. HIUS = high-intensity ultrasound. NIPS = neoadjuvant intraperitoneal and systemic chemotherapy. Hydrogel = using hydrogel as a delivery system for local regions.

**Table 1 jcm-12-00103-t001:** Different studies of HIPEC in different diseases.

Ref.	Disease	Type	Group	Survival	Death and Complication	Recurrence
Glehen [78]	colorectal cancer	retrospective multicenter study	506	overall median survival: 19.2 months complete CRS vs. not complete CRS: 32.4 months vs. 8.4 months (*p* < 0.001)	complication rate: 22.9% death rate: 4%	371 recurrence (73.3% with 158 (41.9%) peritoneal recurrence
Glehen [79]	colorectal cancer	retrospective study	53	median overall survival: 12.8 months CCR-0 vs. CCR-1 vs. CCR-2: 32.9 vs. 12.5 vs. 8.1 (*p* < 0.001)	complication rate: 23% death rate: 4%	–
Kecmanovic [80]	colorectal cancer	retrospective study	18	median overall survival: 15 months	complication: 8	3 live with cancer progress, 3 died of it
Rosa [81]	colorectal cancer	retrospective study	67	median overall survival: 41 months 3-year overall survival: 43%	complication rate: 35.8%	–
François Quénet [82]	colorectal cancer	PRODIGE 7 a multicenter, randomized, open-label, phase 3 trial	133 (CRS plus HIPEC) vs. 132(CRS)	median overall survival: 41.7 months (CRS plus HIPEC) vs. 41.2 months (CRS)		–
Driel [83]	ovarian cancer	a multicenter, open-label, phase 3 trial	123 (Surgery) vs. 122 (Surgery plus HIPEC)	median overall survival: 33.9 months (Surgery) vs. 45.7 months. (Surgery plus HIPEC) 3-year overall: 48% (Surgery) vs. 62% (Surgery plus HIPEC)	Complication: 122 (Surgery) vs. 118 (Surgery plus HIPEC) death: 62% (Surgery) vs. 50% (Surgery plus HIPEC)	recurrence or death: 89% (Surgery) vs. 81% (Surgery plus HIPEC)

CRS: cytoreductive surgery; CCR-0: complete resection; CCR-1: diameter of residual nodules 5 mm or less; CCR-2: diameter of residual nodules more than 5 mm; HIPEC: hyperthermic intraperitoneal chemotherapy.

**Table 2 jcm-12-00103-t002:** New delivery systems using hydrogels as a carrier for the treatment of peritoneal carcinomatosis from different origins.

Hydrogels	Drugs	In Vitro	In Vivo	Highlight
linoleic acid-coupled Pluronic F-127 (Plu-CLA) [111]	Docetaxel	Gastric cancer cells TMK1	peritoneal metastasis from gastric cancer	docetaxel–Plu-CLA synergistically inhibits peritoneal metastasis and prolongs survival in a peritoneal gastric cancer model.
Albumin Hydrogel Hybridized with Paclitaxel-Loaded Red Blood Cell Membrane Nanoparticles [112]	Paclitaxel	Gastric cancer cells	peritoneal metastasis from gastric cancer	the hydrogel possesses good tumor growth suppression properties after a single injection.
PTX/PECT (gel) [113]	PTX	Colorectal cancer cells CT-26	peritoneal metastasis from colorectal cancer	sustained drug concentration at peritoneal levels in combination with drug in the form of nanoparticle contributes to the enhanced anti-tumor efficacy.
HA nanogels [114]	Cisplatin	–	peritoneal metastasis from gastric cancer (MKN45P cells)	led to a significantly decreased number of peritoneal nodules, especially those smaller than 1.0 mm.

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
