# Peer review of "Development of the Peritoneal Metastasis: A Review of Back-Grounds, Mechanisms, Treatments and Prospects"

_jcm, 2022, doi:10.3390/jcm12010103_

Round 1

Reviewer 1 Report

I find the manuscript in this form confusing and unclear.

The scope of the issue exceeds the capabilities of the review article.

After all, each part would deserve a separate section.

For this reason, the essential sections on peritoneal carcinomatosis are missing or simplified into a few words.

It is obvious from the outset that the authors are concerned only with secondary peritoneal metastases, not with primary peritoneal tumors, although they are in many ways the same.

The authors jump from one issue to another - diagnosis, treatment, staging, perspectives, when describing the physiology of the peritoneum they deal with the pathophysiology of peritoneal dissemination, etc.

In this form, I propose to completely revise it.

Reviewer 2 Report

This article summarized in brief and concise manner the peritoneal metastases development and treatment. It is quite clear and useful for a quick reference reading. Paper requires some corrections.

Page 1 unclear paragraph, need revision

 Peritoneal carcinomatosis refers to the development of several separate gastrointestinal 28 and gynecological carcinomas in the peritoneal cavity, and related carcinomas include 29 colorectal carcinoma, gastric carcinoma, ovarian carcinoma, etc.[1].

Page 2 lines 42-55: it is true that CT scan has low sensibility, but on a cost-effective point of view, CT scan is the first choice during follow up detection of peritoneal metastases. Second-level imaging can be proposed in selected patients (e.g. elective surgery). PET-CT has even a lower sensitivity compared to CT for small and mucinous PM and should be proposed in differential diagnosis in selected cases.

Page 3, title “peritoneal carcinomatosis”

Carcinomatosi is an old term that tends to be abandoned in favour of peritoneal metastases, since that term refers to clinical syndrome rather than to stage IV condition (or III in case of ovarian cancer) amenable of potentially curative treatment in patients. Carcinomatosis is not wrong, but I suggest to change it (e.g. Development of PM)

Page 4 line 144: detached cells cannot survive in peritoneal space unless some molecular changes (e.g. adhesion proteins on surface) occur. This can explain why manipulation of T3 tumors do not cause PM in every case

Page 4 line 180

Malignant ascites was a microenvironment in rich of lipid, mostly from differentiated 180 preadipocytes stimulated by cancer cell…. Unclear (IN rich of lipid??)

Line 206: Adhesion to THE peritoneum (the in capital letter? Is it an acronym or just a mistake?)

Round 2

Reviewer 1 Report

Thank you for sending me the corrected manuscript.

I still consider the text to be unnecessarily lengthy, as it covers practically the entire subject of PSM. However, this would deserve a whole book, not a single article.

I therefore leave the decision on publication to the editor-in-chief.